# Molecular Genetic Diversity of Local and Exotic Durum Wheat Genotypes and Their Combining Ability for Agronomic Traits under Water Deficit and Well-Watered Conditions

**DOI:** 10.3390/life13122293

**Published:** 2023-12-01

**Authors:** Ahmed A. Galal, Fatmah A. Safhi, Mahmoud A. El-Hity, Mohamed M. Kamara, Eman M. Gamal El-Din, Medhat Rehan, Mona Farid, Said I. Behiry, Mohamed El-Soda, Elsayed Mansour

**Affiliations:** 1Department of Agronomy, Faculty of Agriculture, Kafrelsheikh University, Kafr El-Sheikh 33516, Egypt; aa.galal79@gmail.com (A.A.G.);; 2Department of Biology, College of Science, Princess Nourah bint Abdulrahman University, Riyadh 11671, Saudi Arabia; faalsafhi@pnu.edu.sa; 3Department of Plant Production and Protection, College of Agriculture and Veterinary Medicine, Qassim University, Burydah 51452, Saudi Arabia; medhat.rehan@agr.kfs.edu.eg; 4Department of Genetics, Faculty of Agriculture, Kafrelsheikh University, Kafr El-Sheikh 33516, Egypt; mfarid_eg2000@yahoo.com; 5Agricultural Botany Department, Faculty of Agriculture (Saba Basha), Alexandria University, Alexandria 21531, Egypt; 6Department of Genetics, Faculty of Agriculture, Cairo University, Giza 12613, Egypt; mohamed.elsoda@agr.cu.edu.eg; 7Department of Crop Science, Faculty of Agriculture, Zagazig University, Zagazig 44519, Egypt; sayed_mansour_84@yahoo.es

**Keywords:** arid environment, biodiversity, cluster analysis, stress, tolerance indices, genetic variation, heterosis, heritability, principal component analysis, sustainability, yield-related traits

## Abstract

Water deficit poses significant environmental stress that adversely affects the growth and productivity of durum wheat. Moreover, projections of climate change suggest an increase in the frequency and severity of droughts, particularly in arid regions. Consequently, there is an urgent need to develop drought-tolerant and high-yielding genotypes to ensure sustained production and global food security in response to population growth. This study aimed to explore the genetic diversity among local and exotic durum wheat genotypes using simple sequence repeat (SSR) markers and, additionally, to explore the combining ability and agronomic performance of assessed durum wheat genotypes and their 28 F_1_ crosses under normal and drought stress conditions. The investigated SSRs highlighted and confirmed the high genetic variation among the evaluated parental durum wheat genotypes. These diverse eight parental genotypes were consequently used to develop 28 F1s through a diallel mating design. The parental durum genotypes and their developed 28 F1s were assessed under normal and drought stress conditions. The evaluated genotypes were analyzed for their general and specific combining abilities as well as heterosis for agronomic traits under both conditions. The local cultivar Bani-Suef-7 (P8) is maintained as an effective combiner for developing shortened genotypes and improving earliness. Moreover, the local cultivars Bani-Suef-5 (P7) and Bani-Suef-7 (P8) along with the exotic line W1520 (P6) demonstrated excellent general combining ability for improving grain yield and its components under drought stress conditions. Furthermore, valuable specific hybrid combinations, W988 × W994 (P1 × P2), W996 × W1518 (P3 × P5), W1011 × W1520 (P4 × P6), and Bani-Suef-5 × Bani-Suef-7 (P7 × P8), were identified for grain yield and its components under drought stress conditions. The assessed 36 genotypes were grouped according to tolerance indices into five clusters varying from highly drought-sensitive genotypes (group E) to highly drought-tolerant (group A). The genotypes in cluster A (two crosses) followed by thirteen crosses in cluster B displayed higher drought tolerance compared to the other crosses and their parental genotypes. Subsequently, these hybrids could be considered valuable candidates in future durum wheat breeding programs to develop desired segregants under water-deficit conditions. Strong positive relationships were observed between grain yield and number of grains per spike, plant height, and 1000-grain weight under water-deficit conditions. These results highlight the significance of these traits for indirect selection under drought stress conditions, particularly in the early stages of breeding, owing to their convenient measurability.

## 1. Introduction

Durum wheat (*Triticum turgidum* L. ssp. durum, Desf.) is a dominant cultivated cereal crop [1]. It is cultivated on an area of approximately 16 million hectares worldwide, with a global production of 38 million tonnes [2]. As a vital food source, it contains carbohydrates, dietary proteins, fiber, calcium, zinc, lipids, and energy [3]. The Mediterranean region takes precedence in durum wheat production and cultivation, with its countries being the largest importers and consumers of durum wheat. Contrary to common bread wheat, grains of durum wheat are harder, exhibit higher yellow pigment content, possess relatively more grain protein, and generally contain non-stretchable gluten [4]. Therefore, most of the durum wheat produced worldwide is used for making denser food. This distinguishes durum wheat as a primary ingredient in the production of the dense and stiff dough of pasta [5]. In North Africa and West Asia, durum wheat is often utilized to make local foods such as freekeh, couscous, and bulgur [6]. Dense durum wheat bread is a staple in the Mediterranean region, owing to its unique texture and taste [7].

Climate change poses a global threat to agricultural production and nutritional security [8,9]. Rising temperatures and precipitation fluctuations are predicted to become more severe and frequent [10]. Durum wheat is commonly grown in Mediterranean regions and arid environments with a top priority of managing water resources [11]. Water stress is a significant abiotic stressor that poses a considerable threat to global durum wheat production [12]. Consequently, one of the main objectives of breeding programs is to improve tolerance and adaptability to drought stress [13,14]. Ongoing efforts focus on developing high-yielding and drought-tolerant durum wheat genotypes, which is necessitated especially in light of current climate change [15]. Notwithstanding, limited knowledge concerning the potentiality of the available genetic material hinders progress in improving drought tolerance [16]. This can be achieved by estimating the combining ability of the available plant materials under drought stress conditions [17]. General and specific combining abilities (GCA and SCA) are biometric analyses that help in selecting suitable parental genotypes for crossing and identifying promising recombinants with improved drought tolerance [18]. Furthermore, diallel mating contributes to explore the heterotic effects at early generations of breeding programs [19,20]. Accordingly, employing diallel mating, GCA and SCA analyses effectively identify promising parents and offspring for targeted traits under drought stress. Previous studies have successfully applied diallel mating to identify superior parents and achieve advancements in stress tolerance traits in durum wheat [21,22,23,24].

The success of breeding programs relies on the presence of substantial genetic variation within populations [25]. Morphological and biochemical markers are valuable indicators of genetic diversity, but their dependability is compromised by their susceptibility to environmental influences [26]. In contrast, DNA markers are more stable, reliable, and reproducible. Simple sequence repeats (SSRs) or microsatellites remain invaluable markers considering their adaptability to simple PCR-based assay and co-dominant transmission [27], relative abundance, multi-allelic nature, genome coverage, and information content [28,29]. Therefore, SSR markers are pivotal for studying genetic differences between genotypes [29]. Exploring the genetic diversity and combining ability of durum wheat genotypes under water deficit is essential for developing drought-tolerant and high-yielding genotypes. We hypothesized that the evaluated durum wheat genotypes would exhibit substantial genetic variation, significantly contributing to the development of promising durum wheat genotypes. Accordingly, the objectives of this study were to (i) investigate genetic distance among local and exotic durum wheat genotypes using SSR markers; (ii) evaluate the agronomic performance of eight local and exotic durum wheat genotypes and their 28 F1 crosses under well-watered and drought stress conditions; (iii) explore GCA, SCA, and heterosis for the evaluated traits; and (iv) study the relationship among measured traits under normal and drought stress conditions.

## 2. Materials and Methods

### 2.1. Molecular Characterization of Parental Genotypes

Eight durum wheat genotypes comprising two local cultivars and six exotic genotypes (CIMMYT lines) were used for this study (Appendix A). These genotypes were selected based on their tolerance to water deficit from a prior screening trial including 25 durum wheat genotypes during the 2020–2021 winter season. DNA extraction was performed from fresh leaves of the parental genotypes using the CTAB method [30]. The consistency and quantity of DNA were assessed utilizing a Nano-Drop spectrophotometer. Fifteen SSR markers were utilized in this study. The applied SSR markers were identified from earlier studies based on their consistent association with drought tolerance in wheat [31,32,33]. The sequences of applied primers are detailed in Table 1.

The polymerase chain reaction was applied utilizing a 10 μL reaction volume containing 1 μL of 20 ng/μL genomic DNA template, 0.2 mM dNTPs, 2 mM MgCl_2_, 1 Taq DNA polymerase unit, and 0.5 μM forward and reverse primers. The reaction was applied by pre-denaturation at 94 °C for 2 min followed by 94 °C for 30 s, annealing at 55–60 °C (based on primer Tm) for 30 s, 30 s of extension at 72 °C for 35 cycles, and ending with 3 min of elongation at 72 °C. Amplification products were analyzed in 1.5% agarose gel. The amplified bands were graded for each SSR marker based on the absence or presence of the bands generating a binary data matrix of (0) and (1). Allele number, gene diversity, major allele frequency, and polymorphic information content (PIC) were determined for all markers using Power-Marker (version 3.25). Genetic distances were determined utilizing the PAST program. The dendrogram tree was performed with the unweighted pair group method employing arithmetic averages (UPGMA) within the computational MVSP package version 3.1.

### 2.2. Hybridization and Field Evaluation

A half-diallel mating design (8 × 8) excluding reciprocals was utilized to produce 21 F1 hybrids during the growing season of 2021–2022. Hand emasculation and pollination were applied to develop grains of twenty-eight hybrids. The assessed durum wheat genotypes and their F1 crosses were evaluated under two irrigation levels at the Experimental Farm, Kafrelsheikh University (31°6′ N, 30°56′ E), Egypt, during the growing season of 2022–2023. The two irrigation regimes were separated by a 6 m wide alley to avoid water leakage. The first treatment (well-watered conditions) was irrigated five times throughout the whole season with a total amount of 4400 m^3^/ha. Otherwise, the second treatment was irrigated twice throughout the season with a total of approximately 2850 m^3^/ha, providing drought stress conditions. The experimental site had an average annual of rainfall 75 mm. The climatic data of the two growing seasons are presented in Appendix A. The soil properties of the experimental site are displayed in Appendix A. The applied experimental design was a Randomized Complete Block Design (RCBD) with three replications for each irrigation treatment. Each genotype was sown in 2 rows 3 m-long, with a 0.30 m space between the rows and a 0.15 m space between the plants. Fertilizers were applied at rates of 180 kg N/ha, 57 kg K_2_O, and 35 kg P_2_O_5_/ha, for nitrogen, potassium, and phosphorus, respectively.

### 2.3. Data Collection

Days to 50% heading was determined as number of days from sowing to date when 50% spikes completely appeared in each plot [34]. At physiological maturity, plant height was measured (in centimeters) as the distance from the soil surface to the spike tip. Spike length (in centimeters) and number of grains per spike were measured from ten spikes that were randomly collected from each plot [35]. The 1000-grain weight (in grams) was assessed as the weight of 1000 grains. Grain yield was estimated by harvesting ten plants from each plot which then were dried, threshed, and finally recorded as grain yield per plant (g).

### 2.4. Drought Tolerance Indices

Four tolerance indices were used to distinguish drought-tolerant and drought-sensitive genotypes. Geometric mean productivity (GMP) was calculated following Fernandez [36] using the following equation $GMP=\sqrt{Y_{s}\times Y_{p}}$. Yield index (YI) was calculated as outlined by Gavuzzi et al. [37] using the following equation, YI = $\frac{Y_{s}}{{Ῡ}_{s}}$. Mean productivity (MP) was calculated according to Rosielle and Hamblin [24] using the following equation $MP=\frac{Y_{s}+Y_{p}}{2}$. Stress tolerance index (STI) was calculated following Fernandez [36] using the following equation $STI=\frac{Y_{s}\times Y_{p}}{{\left({Ȳ}_{p}\right)}^{2}}$ where Y_s_ is the grain yield of each genotype under drought stress conditions, Y_p_ is the grain yield of each genotype under normal conditions, and Ȳ_s_ and Ȳ_p_ are the means of all evaluated genotypes under drought stress and normal conditions, respectively.

### 2.5. Statistical Analysis

The analysis of variance was implemented for all studied traits and the least significant difference (*p* < 0.05) test was employed to determine the significance of variations among means. Griffing’s (1956) method 2 model 1 [38] was applied for combining ability (GCA and SCA) analysis using the following model: xij = µ + ĝi + ĝj + ŝij + eijkl, where xij is the recorded value of the cross between parent (i) and parent (j), µ is the population mean, ĝi and ĝj are the GCA effect ith and jth parents, ŝij is the SCA effect for the cross between (i) and (j) parent, and eijkl is the environmental effect peculiar to the jkl observation. The GCA/SCA ratio was calculated by comparing the mean squares of GCA and SCA. The difference in GCA and SCA was computed by considering the standard error and the tabulated t-value. Heterosis relative to better parent was computed as follows: better-parent (BP) heterosis = (F1 – BP)/BP × 100. The principal component analysis (PCA) and heatmap with clustering were performed utilizing averages of the studied traits to explore their relationships. All performed analyses were implemented utilizing R statistical software (version 4.1.1).

## 3. Results

### 3.1. Molecular Diversity among Evaluated Parental Genotypes

Fifteen microsatellite markers were utilized to differentiate the genetic diversity among the tested parental genotypes. A total of 59 polymorphic alleles were determined. The allele number per locus ranged from 2 (Xgwm 99) to 6 (Xgwm 357 and Xwmc 78), with an average of 3.93 alleles per locus (Table 2 and Appendix A). The major allele frequency presented an average of 0.43, ranging from 0.25 to 0.81. Heterozygosity (He) differed from 0.33 to 0.82, with an average of 0.59. The lowermost and uppermost He values were observed in the Xgwm 108 and Xwmc 78 markers, respectively. Moreover, the polymorphic information content (PIC) had an average of 0.58, with a range between 0.32 (Xgwm 108) and 0.80 (Xwmc 78).

The genetic distance determined by SSR markers differed from 0.08 to 0.46, with an average of 0.23 (Table 3). The uppermost genetic distance was detected between P_1_ and P_5_ (0.46). On the other hand, the lowest genetic distance was determined between the two parental lines P4 and P8 (0.08). The dendrogram constructed from the genetic distance matrix separated the genotypes into three main clusters, with internal sub-clusters exhibiting contrasting degrees of diversity (Figure 1). Group 1 included P5. Group II comprised seven genotypes, which were divided into two sub-groups. The first sub-group included P_1_, and the second sub-group retained five genotypes, which were subdivided into two sub-sub clusters. The first one comprised P3 and P7, while P4, P6, and P8 formed the second one.

### 3.2. Analysis of Variance 

Significant variations were observed among the evaluated parents, F1 crosses, and parents vs. crosses for most studied traits under normal and stressed conditions (Table 4). These significant differences among the evaluated genotypes indicate sufficient variability that could be utilized for breeding drought-tolerant genotypes. The mean squares of GCA and SCA were highly significant for all studied traits, except the number of spikelets/spike under water-deficit conditions. The ratio of GCA to SCA was less than the unity for all studied traits, except day to heading and number of spikelets/spike under both conditions and 1000-grain weight under well-watered conditions.

### 3.3. Performance of the Evaluated Parental Genotypes and F_1_ Combinations

The performance of evaluated parental genotypes and their F1 crosses displayed great variations for all evaluated traits under well-watered and stressed conditions. In response to drought stress, all evaluated genotypes exhibited an acceleration in heading by 6.54 days compared to well-watered conditions. The parent P5 and the hybrid P4 × P8 exhibited the earliest heading, while P2 and the hybrid P2 × P3 showed the latest heading under both treatments (Figure 2A). Drought stress resulted in a significant reduction in plant height by 7.5% compared to normal conditions. The parent P7 and hybrid P1 × P2 had the tallest plants, while the shortest plant height was assigned for P5 and P2 × P8, P5 × P8, and P5 × P6 under drought stress and normal conditions (Figure 2B). Likewise, spike length decreased by 25.83% under drought stress. The parental genotype P7 and hybrids P3 × P4 and P2 × P4 exhibited the longest spike under both treatments (Figure 2C). Likewise, the number of spikelets/spike was reduced by 18.25% due to water-deficit conditions. The parent P8 and the crosses P2 × P3, P2 × P8, and P2 × P7 recorded the highest values of spikelet number per spike under both irrigation treatments (Figure 2D). The number of spikes per plant was reduced by 17.8% due to drought stress. The highest values were assigned for the parents P7 and P8 and the hybrids P5 × P8, P1 × P8, P4 × P7, and P2 × P8 under stressed and non-stressed conditions (Figure 3A). Likewise, the number of grains per spike significantly decreased by 13% under water deficit (Figure 3B). The parents P7, P8, and P6 and crosses P4 × P7, P6 × P8, and P2 × P7 showed the greatest values under normal and stressed conditions. The 1000-grain weight was substantially impacted by water deficit; it declined by 12.5% compared to normal conditions. The parents P7 and P8 and P6 had the heaviest grains; the crosses P7 × P8, P1 × P8, and P4 × P7 exhibited the heaviest weight under both treatments (Figure 3C). The grain yield was destructively influenced by water deficit; it suffered an 18.4% reduction under water-deficit conditions. The parental genotypes P6, P7, and P8 and the crosses P7 × P8, P6 × P7, P4 × P6, P2 × P7, P4 × P8, and P6 × P8 produced the uppermost grain yield under normal and water-deficit conditions (Figure 3D).

### 3.4. Classification of Evaluated Genotypes

The evaluated thirty-six genotypes were clustered into five distinct groups based on their tolerance to drought stress (Figure 4). Cluster (A) comprised two genotypes; namely P7 × P8 and P6 × P8, demonstrating the highest tolerance indices. This identifies these genotypes as highly drought tolerant. Cluster (B) included fifteen genotypes that displayed high values; accordingly, they could be categorized as drought-tolerant genotypes. Likewise, cluster (C) contained six genotypes (P1 × P7, P1 × P2, P3 × P5, P6, P2 × P4, and P4 × P5) with intermediate tolerance indices; therefore, they are classified as moderately drought-tolerant genotypes. On the other hand, cluster (E) with four genotypes and (D) with nine genotypes displayed the lowest values of the tolerance indices, respectively. Therefore, they are considered highly drought-sensitive and sensitive genotypes, respectively.

### 3.5. General Combining Ability (GCA) Effects

Positive and significant GCA effects are desirable for all the traits examined, except for plant height and days to heading, where the preference is for negative values. The parental line P1 was found to be an undesirable combiner for most studied traits; it exhibited negative (significant or insignificant) effects (Table 5). The parent line P2 gave highly significant or significant positive effects for spike length and number of spikelets per spike under both conditions. P3 displayed highly significant effects for spike length under normal irrigation. Otherwise, it displayed insignificant or undesirable effects for other traits. The parental line P4 expressed significant negative effects for days to heading under normal irrigation and highly significant desirable effects for spike length under well-watered conditions (Table 5). The parental line P5 indicated negative and significant effects for days to heading under normal conditions and highly significant effects for plant height under drought and normal conditions. The parental line P6 gave significant negative effects for days to heading under both conditions. Otherwise, it exhibited desirable significant effects for number of spikelets per spike under normal irrigation, number of grains per spike, and grain yield per plant under drought and normal conditions. The parental cultivar P7 expressed highly significant negative effects for days to heading under normal and drought conditions, indicating that this parent could be considered an excellent combiner for improving earliness in durum wheat. Additionally, this parent could be an excellent general combiner for grain yield and its components. Moreover, the parental cultivar P8 appeared to be an excellent general combiner for short plant type and earliness, as well as the number of spikelets, spike length, number of grains per spike, number of spikes per plant, 1000-grain weight, and grain yield per plant under both normal and stressed conditions.

### 3.6. Specific Combining Ability (SCA) Estimates 

The hybrids displayed diverse variations in the observed GCAs for all studied traits. Under well-watered conditions, the crosses P4 × P8, P3 × P5, and P3 × P7 expressed significant negative (S̑ij) effects for days to heading (Table 6), while under stress conditions the crosses P1 × P2, P1 × P3, P2 × P8, P3 × P5, and P4 × P8 showed significant negative (S̑ij) effects. These crosses could be employed in breeding programs of durum wheat to enhance earliness. For plant height, the data showed that two crosses (P1 × P7) and (P3 × P4) expressed significant negative (S̑ij) effects under both environments, indicating that these genotypes could be used as excellent combiners for breeding short-stature genotypes under drought-stress conditions. Regarding spike length, five and six crosses had highly significant positive (S̑ij) effects under drought and well-watered conditions. The highest desirable (S̑ij) effects were detected by the crosses P5 × P7 and P5 × P8 under both conditions. For the number of spikelets per spike, the crosses P2 × P4, P2 × P8, P3 × P4, and P3 × P6 demonstrated the maximum positive SCA values under well-watered conditions, while the cross P2 × P3 recorded the highest positive SCA value under drought stress conditions.

The crosses P1 × P8, P4 × P5, and P5 × P8 showed positive and significant effects for the number of spikes per plant under both conditions. Regarding the number of grains per spike, five and nine crosses displayed significant and positive (S̑ij) effects under well-watered and stressed conditions, in the same order. The uppermost desirable (S̑ij) effects were assigned to cross P1 × P2, followed by P4 × P7 and P5 × P8 under both treatments. In addition, the crosses P2 × P7 and P5 × P7 possessed the highest positive effects for 1000-grain weight under normal irrigation, while P1 × P2, P2 × P6, P2 × P7, P4 × P6, P4 × P7, and P5 × P7 under stress conditions displayed the highest desirable (S̑ij) effects. Significant and positive SCA effects for grain yield per plant were obtained by twelve crosses under both irrigation treatments. The greatest significant positive SCA effects were recorded by P1 × P2, P3 × P5, P4 × P6, and P7 × P8 under both normal and stressed conditions.

### 3.7. Interrelationship among Evaluated Traits

Principal component analysis was applied to explore the association among studied traits under drought stress. The first two principal components explained most of the variability, at 68.5% (48.6% by PCA1 and 19.9% by PCA2). Therefore, the first two PCAs were employed to perform the PC biplot (Figure 5). PCA1 divided the genotypes into positive and negative sides of PCA1. The studied agronomic traits were associated with the genotypes on the positive side of PCA1. This indicates that the genotypes located on the positive side of PCA1 had high agronomic performance, particularly P7 × P8, P6 × P8, P7, and P8. On the contrary, the genotypes are on the opposite side of PC1, possessing lower agronomic performance, especially P5 and P1 × P5. Strong positive correlation was observed between the grain yield and each of spike length, plant height, number of grains per spike, and 1000-grain weight. On the other hand, a negative relationship was detected between yield traits and days to heading. Likewise, the heatmap and hierarchical clustering based on the studied traits separated the evaluated genotypes into different clusters (Figure 6). Moreover, a heatmap analysis exhibited the association between the evaluated genotypes and the studied traits using a color scale under drought stress. The blue color displays high values of measured agronomic traits, whereas the red color exhibits low values. The genotypes P6 P7, P8, P7 × P8, and P6 × P8 displayed superior values for all agronomic traits (represented in blue). Instead, P1, P5, P1 × P4, and P5 × P6 possessed the lowest values (red values) under water-deficit conditions.

### 3.8. Heterosis Relative to Better Parent

Significant better-parent heterosis (BPH) in both positive and negative directions was observed for all studied traits (Appendix A). Fifteen cross combinations under normal irrigation and stressed conditions expressed negative and significant heterotic values towards earliness. The highest negative values were estimated for the crosses P1 × P2, P2 × P8, P1 × P3, P3 × P5, and P3 × P7 under both conditions. Similarly, five cross combinations P1 × P7, P2 × P8, P5 × P7, P6 × P7, and P7 × P8 showed desirable heterotic effects for plant height under well-watered and stressed conditions. The maximum positive and significant BPH for spike length was recorded by P1 × P2, P2 × P3, and P2 × P4 under well-watered and stressed conditions. Only P2 × P3 showed significant positive heterosis for the number of spikelets/spike under water-deficit conditions. The uppermost positive significant heterotic effects for the number of spikes/plant were recorded by P1 × P5, P3 × P5, P4 × P5, and P7 × P8 under both treatments. The highest positive and significant heterotic effects for the number of grains/spike was exhibited by P3 × P5 and P6 × P8. Regarding 1000-grain weight, the crosses P1 × P2, P2 × P3, and P4 × P5 had superior heterotic effects under normal and stressed conditions. Twenty-three and twelve hybrid combinations had superior heterotic effects for grain yield/plant under normal and stressed conditions. Out of these hybrids, P1 × P5, P7 × P8, P6 × P8, P4 × P6, and P1 × P2 exhibited the highest heterosis under both conditions.

## 4. Discussion

Developing drought-tolerant and high-yielding durum wheat genotypes is critical for ensuring food security, especially under abrupt climate fluctuations and a growing worldwide population. Exploring genetic diversity and combining abilities are essential for improving drought tolerance in durum wheat. The significant difference detected among the evaluated genotypes indicates sufficient variability and promising resources that could be utilized for breeding drought-tolerant genotypes. In this context, Nouri et al. [39]; Mohammadi [40]; Salsman et al. [41]; and Mohammadi et al. [42] reported high genetic diversity for agronomic traits in durum wheat under drought stress conditions. 

Molecular markers were proven to be a successful tool for exploring genetic variability and diversity among the evaluated genotypes [43,44]. The SSRs are useful DNA-based markers for studying genetic variation within durum wheat genotypes [45,46]. In the present study, SSR markers demonstrated considerable levels of genetic diversity among the evaluated parental durum genotypes. Likewise, Eujayl et al. [47]; Maccaferri et al. [48]; Dagnaw et al. [46]; Marzario et al. [49]; Dukamo et al. [50]; and Almarri et al. [51] demonstrated that genetic diversity among assessed genotypes could be employed in breeding programs of durum wheat. 

The crossing among diverse parental genotypes could lead to the accumulation of favorable alleles and result in superior hybrids. The applied SSR markers classified the evaluated parental genotypes into different groups. Accordingly, the developed hybrid from crossing P6 and P7, which were from different groups, produced greater grain yield under drought stress and normal conditions. This demonstrates there is a potential to develop more effective hybrids with enhanced drought tolerance by crossing different genotypes from diverse clusters. Hence, the obtained results demonstrated that SSR markers can potentially be employed to explore genetic diversity and provide guidelines for parental selection in durum wheat breeding programs. This is highly beneficial in durum wheat breeding in reducing the crosses which need to be assessed under field conditions. Furthermore, the applied markers Xwmc 78, Xgwm 337, and Xwmc 89 had higher PIC values >0.65. This proves the effectiveness of these three loci in discriminating diverse durum wheat genotypes. The used SSR markers did not separate the evaluated genotypes based on geographic location. In this respect, Asmamaw et al. [52] disclosed that it is not necessarily that all genotypes derived from the same region are located in the same cluster. This revealed that the genetic variations in durum wheat are not completely associated with geographic distribution. A water deficit caused significant reductions in all studied traits relative to well-watered conditions. The reduction in plant height might be caused by the limited water uptake from the root system and, consequently, reduced cell division and elongation. The remarkable decreases in yield traits under drought stress could be caused by the shortened grain-filling period, which subsequently decreased the 1000-grain weight and grain yield per plant. Likewise, Zhao et al. [53]; Hafsi et al. [54]; Sukumaran et al. [55]; Giunta et al. [56]; Pour-Aboughadareh et al. [12]; and Shirvani et al. [57] elucidated a significant reduction in the grain yield and genotypic differences of durum wheat under drought stress. The evaluated parental genotypes and their derived hybrid combinations were classified based on their tolerance to water deficit into five categories (A-E) differing from highly tolerant to highly sensitive genotypes. The crosses P7 × P8 and P6 × P8 were determined to be drought tolerant. These genotypes exhibited superior agronomic performance compared to the sensitive genotypes. Accordingly, these tolerant genotypes could be exploited in breeding programs of durum wheat to improve grain yield under drought stress conditions. In this context, several published studies applied cluster analysis and tolerance indices to classify the genotypes under drought stress conditions [50,58,59,60,61].

Breeders preferred the positive SCA effects for all agronomic traits, except plant height and days to heading for which negative values are favored. The GCA effects of the parental genotypes varied from positive to negative under both irrigation regimes. The present study identified the parental line P8 as an effective combiner for shortened plant height and earliness under both irrigation treatments. This indicated that this parent could be beneficial in developing dwarf and early maturity genotypes. Early heading and short plant height could be considered as an escape mechanism and a resilient adaptation to avoid terminal drought stress. Furthermore, the local parental genotypes P7 and P8 and the exotic line P6 were identified with highly favorable GCA for grain yield and its components under drought stress conditions. These genotypes could be considered as potential genetic materials to improve the yielding ability of durum wheat under drought stress. These genotypes could be utilized to transfer their favorable alleles to progenies for enhancing grain yield under water-limited conditions. Furthermore, such genotypes could combine effectively with other genotypes to produce superior progeny. The corresponding findings were disclosed by Shamuyarira et al. [62]; Kamara et al. [63]; and Shamuyarira et al. [62].

The observed SCA effects in all developed hybrids indicated a significant improvement in at least one trait. Hybrids that exhibited significant SCA effects are ideal candidates for selecting transgressive segregates. The hybrid combinations P1 × P2, P3 × P5, P4 × P6, and P7 × P8 can be identified as specific combiners for enhancing grain yield and its related traits. Therefore, these hybrids offer the potential for obtaining desirable recombinants and could be utilized to enhance heterosis and develop high-yielding genotypes under drought stress conditions. It is noteworthy that these hybrids were obtained by crossing parents with either good × poor or good × good general combiners. This could be due to one parent acting as a strong combiner with positive additive effects, whereas the other parent with poor GCA contributes to epistatic effects. These results are in agreement with those of Askander [64]; Semahegn et al. [65]; Sharma et al. [66]; and Mwadzingeni et al. [67]. Additionally, the two hybrids P4 × P6 and P7 × P8 exhibited highly significant positive SCA coupled with high heterotic effects for grain yield and yield attributes. Accordingly, promising segregants might be predicted from these hybrids. Therefore, these hybrids might be effectively used to enhance these characteristics in both optimal and drought-stressed environments [18].

## 5. Conclusions

The applied SSR markers indicated high genetic variation among the assessed parental durum wheat genotypes. The obtained results of the molecular diversity analysis could be beneficial in durum wheat breeding programs to identify distinct genotypes for crossing. The local cultivar Bani-Suef-7 (P8) was identified as an effective combiner for developing shortened genotypes and improving earliness. Additionally, the local cultivars Bani-Suef-5 (P7) and Bani-Suef-7 (P8), as well as the exotic line W1520 (P6), were recognized as excellent combiners to improve grain yield and its components under drought stress conditions. Furthermore, the hybrid combinations W988 × W994 (P1 × P2), W996 × W1518 (P3 × P5), W1011 × W1520 (P4 × P6), and Bani-Suef-5 × Bani-Suef-7 (P7 × P8) were identified as valuable specific combiners for grain yield and its components under drought stress conditions. Consequently, the tolerant hybrids could be employed to develop desired segregants and improve durum wheat under water-deficit conditions.

## Figures and Tables

**Figure 1 life-13-02293-f001:**
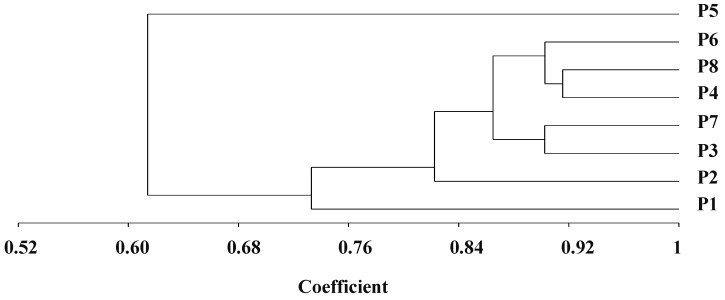
Dendrogram constructed from UPGMA cluster analysis of eight parental durum wheat genotypes according to SSR markers.

**Figure 2 life-13-02293-f002:**
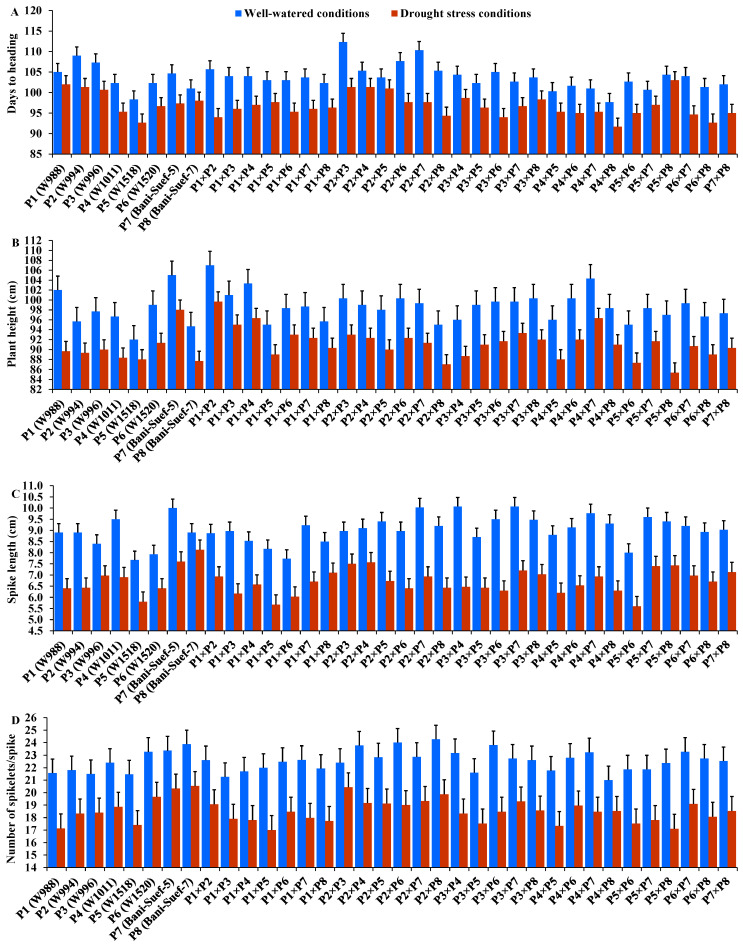
Performance of the 28 durum wheat F1s and their parental genotypes for days to heading (**A**); plant height (**B**); spike length (**C**); and number of spikelets per spike (**D**). The bars on the top of the columns represent LSD (*p* < 0.05).

**Figure 3 life-13-02293-f003:**
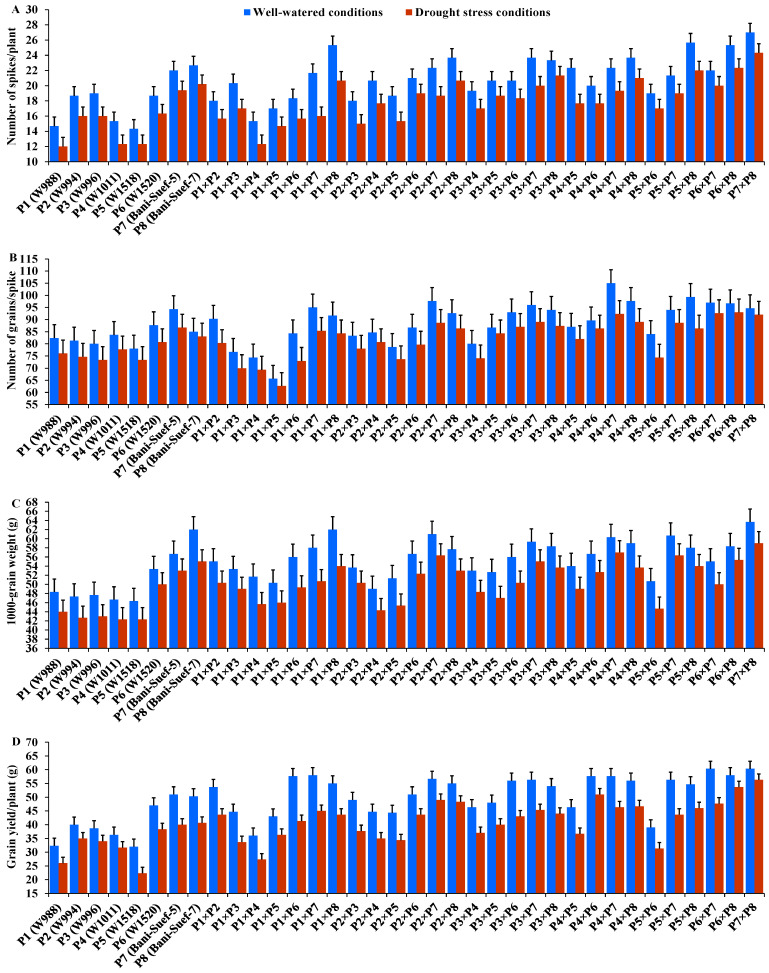
Performance of the 28 durum wheat F1s and their parental genotypes for number of spikes/plant (**A**); number of grains/spike (**B**); 1000-grain weight (**C**); and grain yield/plant (**D**). The bars on the top of the columns represent LSD (*p* < 0.05).

**Figure 4 life-13-02293-f004:**
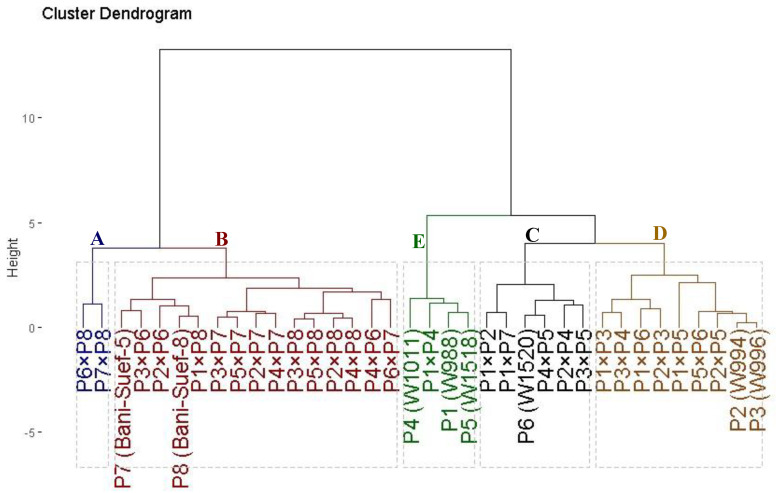
Dendrogram of thirty-six durum wheat genotypes (eight parental genotypes and their twenty-eight F1s) based on four drought tolerance indices (MP, GM, YI, and STI). The genotypes were grouped into five clusters varying from highly drought-sensitive genotypes (group E) to highly drought-tolerant (group A).

**Figure 5 life-13-02293-f005:**
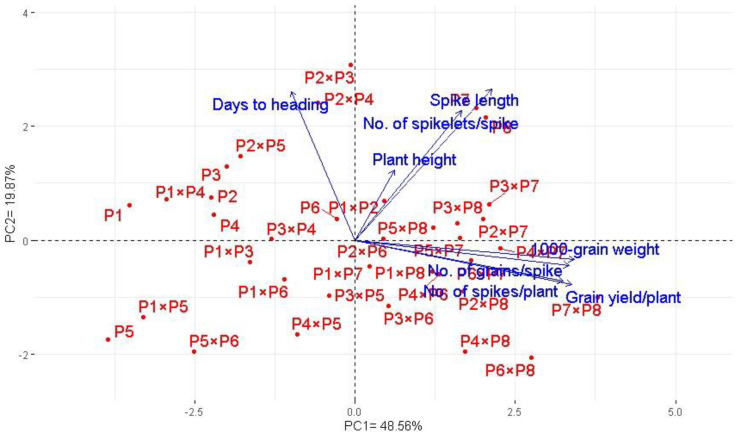
Principal component biplot for the evaluated 28 durum wheat F1s and their parental genotypes based on the studied traits under drought stress conditions.

**Figure 6 life-13-02293-f006:**
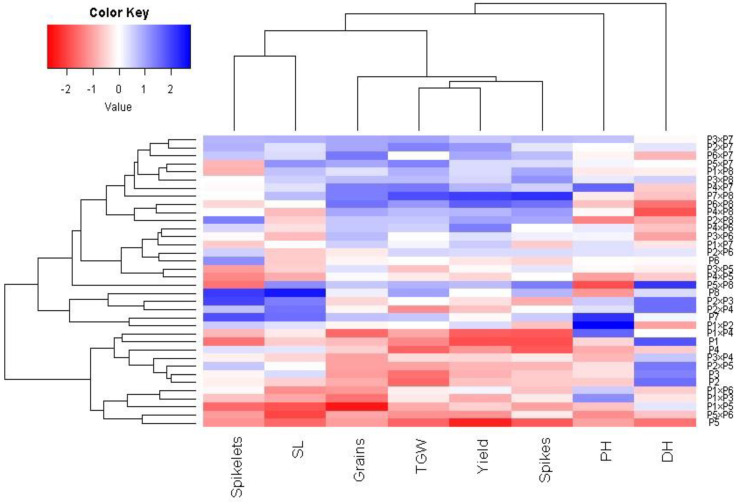
Heatmap and hierarchical clustering dividing the assessed 28 durum wheat F1s and their parental genotypes under drought stress into different clusters according to studied agronomic traits. PH: plant height, HD: days to heading, SL: spike length, spikelets: No. of spikelets per spike, Spikes: No. of spikes per plant, Grain: No. of grain/spike, TGW: 1000-grain weight, and Yield: Grain yield per plant.

**Table 1 life-13-02293-t001:** The applied simple sequence repeat (SSR) markers.

Marker	Forward Primer	Reverse Primer
Xgwm 11	GGATAGTCAGACAATTCTTGTG	GTGAATTGTGTCTTGTATGCTTCC
Xgwm 99	AAGATGGACGTATGCATCACA	GCCATATTTGATGACGCATA
Xgwm 108	CGACAATGGGGTCTTAGCAT	TGCACACTTAAATTACATCCGC
Xgwm 186	GCAGAGCCTGGTTCAAAAAG	CGCCTCTAGCGAGAGCTATG
Xgwm 337	CCTCTTCCTCCCTCACTTAGC	TGCTAACTGGCCTTTGCC
Xgwm 357	TATGGTCAAAGTTGGACCTCG	AGGCTGCAGCTCTTCTTCAG
Xgwm 389	ATCATGTCGATCTCCTTGACG	TGCCATGCACATTAGCAGAT
Xgwm 484	ACATCGCTCTTCACAAACCC	AGTTCCGGTCATGGCTAGG
Xgwm 603	ACAAACGGTGACAATGCAAGGA	CGCCTCTCTCGTAAGCCTCAAC
Xgwm 626	GATCTAAAATGTTATTTTCTCTC	TGACTATCAGCTAAACGTGT
Xpsp 3200	GTTCTGAAGACATTACGGATG	GAGAATAGCTGGTTTTGTGG
Xwmc 78	AGTAAATCCTCCCTTCGGCTTC	AGCTTCTTTGCTAGTCCGTTGC
Xwmc 89	ATGTCCACGTGCTAGGGAGGTA	TTGCCTCCCAAGACGAAATAAC
Xwmc 118	AGAATTAGCCCTTGAGTTGGTC	CTCCCATCGCTAAAGATGGTAT
Xwmc 304	CGATACAAGGAAGACCAGCC	GGTTCGTCTGGTTCGCAAGT

**Table 2 life-13-02293-t002:** Description of the studied fifteen SSR markers utilized in this study.

Locus	Size Range of Alleles (bp)		Alleles Number	Major Allele Frequency	Gene Diversity (He)	PIC
	Min Allele	Max Allele				
Xgwm 11	50	170	3	0.48	0.60	0.51
Xgwm 99	230	250	2	0.39	0.47	0.36
Xgwm 108	50	400	5	0.81	0.33	0.32
Xgwm 186	50	150	3	0.56	0.54	0.45
Xgwm 337	50	350	4	0.30	0.75	0.70
Xgwm 357	50	600	6	0.38	0.75	0.71
Xgwm 389	130	400	2	0.63	0.47	0.36
Xgwm 484	150	600	4	0.34	0.71	0.65
Xgwm 603	50	900	5	0.33	0.73	0.68
Xgwm 626	50	150	3	0.35	0.66	0.59
Xpsp 3200	70	800	4	0.48	0.65	0.59
Xwmc 78	100	600	6	0.25	0.82	0.80
Xwmc 89	50	200	4	0.34	0.74	0.69
Xwmc 118	40	200	4	0.42	0.68	0.63
Xwmc 118	50	200	4	0.33	0.72	0.67

**Table 3 life-13-02293-t003:** Genetic distance among the evaluated genotypes based on applied SSR markers.

Parent	P1 (W988)	P2 (W994)	P3 (W996)	P4 (W1011)	P5 (W1518)	P6 (W1520)	P7 (Bani-Suef-5)
P_1_ (W988)	-						
P_2_ (W994)	0.24	-					
P_3_ (W996)	0.24	0.15	-				
P_4_ (W1011)	0.31	0.20	0.16	-			
P_5_ (W1518)	0.46	0.39	0.37	0.38	-		
P_6_ (W1520)	0.24	0.17	0.13	0.09	0.40	-	
P_7_ (Bani-Suef-5)	0.23	0.19	0.10	0.13	0.37	0.12	-
P_8_ (Bani-Suef-7)	0.34	0.18	0.13	0.08	0.33	0.11	0.15

**Table 4 life-13-02293-t004:** Mean squares and combining abilities for evaluated agronomic traits under normal (Well-W.) and stressed (Drought-S.) conditions.

SOV	df	Days to Heading		Plant Height (cm)		Spike Length (cm)		No. of Spikelets Per Spike	
		Well-W.	Drought-S.	Well-W.	Drought-S.	Well-W.	Drought-S.	Well-W.	Drought-S.
Replication	2	16.03 **	7.95	150.25 **	88.95 **	3.88 **	5.27 **	11.76 **	6.89 **
Genotype (G)	35	26.89 **	20.29 **	29.28 **	30.59 **	1.18 **	0.80 **	2.10 **	2.65 **
Parent (P)	7	35.88 **	20.52 **	51.52 **	36.93 **	1.78 **	0.93 **	2.82 **	4.81 **
F1 Cross (C)	27	25.55 **	19.35 **	23.86 **	29.27 **	1.00 **	0.80 **	1.97 **	2.08 **
P vs. C	1	0.02	44.37 **	20.02	21.91	1.90 **	0.01	0.54	2.79
Error	70	2.98	3.13	8.79	6.00	0.06	0.07	0.77	0.70
GCA	7	32.77 **	40.07 **	22.92 **	26.48 **	1.19 **	0.69 **	2.59 **	2.61 **
SCA	28	3.01 **	5.94 **	6.47 **	6.13 **	0.19 **	0.16 **	0.48 *	0.45
Error	70	0.99	1.04	2.93	2.00	0.02	0.02	0.26	0.23
K2GCA/K2SCA		1.58	1.18	0.56	0.59	0.67	0.48	1.06	1.10
**SOV**	**df**	**No. of Spikes** **Per Plant**		**No. of Grains** **Per Spike**		**1000-Grain Weight (g)**		**Grain Yield Per** **Plant (g)**	
		**Well-W.**	**Drought-S.**	**Well-W.**	**Drought-S.**	**Well-W.**	**Drought-S.**	**Well-W.**	**Drought-S.**
Replication	2	62.69 **	68.18 **	439.45 **	203.06 **	303.45 **	246.78 **	101.73 **	122.84 **
Genotype (G)	35	29.67 **	28.24 **	205.62 **	172.59 **	67.08 **	65.65 **	202.06 **	175.29 **
Parent (P)	7	30.48 **	35.79 **	78.52 **	71.05 **	98.90 **	83.42 **	174.61 **	130.19 **
F1 Cross (C)	27	23.73 **	22.45 **	228.11 **	192.26 **	43.49 **	48.68 **	132.56 **	138.51 **
P vs. C	1	184.38 **	131.56 **	488.02 **	352.45 **	481.22 **	399.29 **	2270.91 **	1484.13 **
Error	70	5.26	2.68	21.52	11.43	12.81	7.88	2.93	1.77
GCA	7	31.69 **	34.80 **	183.54 **	178.15 **	70.68 **	67.95 **	166.40 **	158.74 **
SCA	28	4.44 **	3.06 **	39.79 **	27.38 **	10.28 **	10.36 **	42.59 **	33.35 **
Error	70	1.75	0.89	7.17	3.81	4.27	2.63	0.98	0.59
K2GCA/K2SCA		1.11	1.56	0.54	0.74	1.10	0.84	0.40	0.48

* and ** imply significance at 0.05 and 0.01 levels of probability, in the same order.

**Table 5 life-13-02293-t005:** Estimates of the general combining ability of eight durum wheat parental genotypes for evaluated agronomic traits under normal (Well-W.) and drought stress (Drought-S.) conditions.

Parent	Days to Heading		Plant Height (cm)		Spike Length (cm)		Number of Spikelets/Spike	
	Well-W.	Drought-S.	Well-W.	Drought-S.	Well-W.	Drought-S.	Well-W.	Drought-S.
P1 (W988)	0.22	0.48	1.52 **	1.47 **	−0.34 **	−0.24 **	−0.51 **	−0.66 **
P2 (W994)	3.48 **	1.54 **	0.26	0.30	0.11 **	0.12 *	0.35 *	0.59 **
P3 (W996)	1.55 **	0.81 **	0.36	0.93 *	0.13 **	0.08	−0.23	0.05
P4 (W1011)	−1.45 **	−0.56	0.29	0.09	0.25 **	0.01	−0.06	−0.05
P5 (W1518)	−1.98 **	0.21	−2.54 **	−2.50 **	−0.38 **	−0.32 **	−0.56 **	−0.86 **
P6 (W1520)	−1.35 **	−1.69 **	−0.01	−0.27	−0.39 **	−0.29 **	0.46 **	0.21
P7 (P7)	1.02 **	−0.86 **	1.93 **	2.17 **	0.57 **	0.43 **	0.30 *	0.44 **
P8 (P8)	−1.48 **	−1.03 **	−1.81 **	−2.10 **	0.14	0.21 **	0.23	0.27
LSD (0.05) gi	0.59	0.60	1.01	0.83	0.08	0.09	0.30	0.28
LSD (0.01) gi	0.78	0.80	1.34	1.10	0.11	0.12	0.39	0.38
**Parent**	**No. of Spikes** **Per Plant**		**No. of Grains** **Per Spike**		**1000-Grain** **Weight (g)**		**Grain Yield** **Per Plant (g)**	
	**Well-W.**	**Drought-S.**	**Well-W.**	**Drought-S.**	**Well-W.**	**Drought-S.**	**Well-W.**	**Drought-S.**
P1 (W988)	−2.02 **	−2.43 **	−4.95 **	−5.69 **	−1.19	−1.83 **	−3.32 **	−4.09 **
P2 (W994)	−0.58	−0.63 *	−1.55	−1.72 **	−1.59 *	−1.39 **	−1.15 **	−0.23
P3 (W996)	−0.15	−0.10	−2.25 **	−1.76 **	−1.33 *	−1.16 *	−1.42 **	−1.53 **
P4 (W1011)	−1.12 **	−1.30 **	−0.65	−0.49	−1.79 **	−1.59 **	−2.85 **	−2.06 **
P5 (W1518)	−1.22 **	−1.13 **	−4.08 **	−3.52 **	−2.46 **	−2.43 **	−5.02 **	−5.09 **
P6 (W1520)	−0.18	0.37	2.45 **	1.64 **	0.11	0.34	2.78 **	2.44 **
P7 (Bani-Suef−5)	1.88 **	1.83 **	7.58 **	6.81 **	3.64 **	3.91 **	6.18 **	4.94 **
P8 (P8)	3.38 **	3.40 **	4.45 **	5.04 **	4.61 **	4.14 **	4.78 **	5.61 **
LSD (0.05) gi	0.78	0.56	1.58	1.15	1.22	0.95	0.58	0.45
LSD (0.01) gi	1.03	0.74	2.09	1.52	1.61	1.27	0.77	0.60

* and ** indicate *p*-value < 0.05 and 0.01, respectively.

**Table 6 life-13-02293-t006:** Estimates of specific combining ability (S̑ij) effects of the 28 F1 crosses for evaluated agronomic traits under well-watered (Well-W.) and water-deficit (Drought-S.) conditions.

Cross	Days to Heading		Plant Height (cm)		Spike Length (cm)		No. of Spikelets/ Spike	
	Well-W.	Drought-S.	Well-W.	Drought-S.	Well-W.	Drought-S.	Well-W.	Drought-S.
P1 × P2 (W988 × W994)	−1.76 **	−5.19 **	6.58 **	6.64 **	0.07	0.37 *	0.22	0.61
P1 × P3 (W988 × W996)	−1.49	−2.46 **	0.48	2.34	0.15	−0.37 *	−0.53	−0.03
P1 × P4 (W988 × W1011)	1.51	−0.09	2.88	3.61 **	−0.40 **	0.10	−0.27	−0.03
P1 × P5 (W988 × W1518)	1.04	−0.19	−2.62	−1.23	−0.13	−0.47 **	0.53	−0.02
P1 × P6 (W988 × W1520)	−0.59	−0.63	−1.82	0.54	−0.56 **	−0.13	−0.02	0.38
P1 × P7 (W988 × Bani-Suef-5)	−0.29	−0.89	−3.42 *	−2.56 *	−0.02	−0.18	0.30	−0.34
P1 × P8 (W988 × Bani-Suef-7)	−0.12	1.71	−2.69	−0.29	−0.22	0.31 *	−0.33	−0.41
P2 × P3 (W994 × W996)	3.58 **	1.81	1.08	0.51	−0.30 *	0.61 **	−0.26	1.26 **
P2 × P4 (W994 × W1011)	−0.42	3.17 **	−0.19	0.77	−0.28 *	0.75 **	0.94 *	0.09
P2 × P5 (W994 × W1518)	−1.56	2.07 *	1.64	0.94	0.65 **	0.24	0.51	0.87
P2 × P6 (W994 × W1520)	0.81	0.64	1.44	1.04	0.22	−0.12	0.65	−0.33
P2 × P7 (W994 × Bani-Suef-5)	3.11 **	−0.29	−1.49	−2.39	0.33 *	−0.30 *	−0.32	−0.22
P2 × P8 (W994 × Bani-Suef-7)	−0.39	−4.36 **	−2.09	−2.46	0.02	−0.58 **	1.15 *	0.48
P3 × P4 (W996 × W1011)	0.51	1.24	−3.29 *	−3.53 **	0.66 **	−0.32 *	0.92 *	−0.21
P3 × P5 (W996 × W1518)	−2.96 **	−2.86 **	2.54	1.31	−0.07	−0.02	−0.15	−0.20
P3 × P6 (W996 × W1520)	0.08	−2.29 *	0.68	−0.26	0.73 **	−0.18	1.03 *	−0.33
P3 × P7 (W996 × Bani-Suef-5)	−2.62 **	−0.56	−1.26	−0.03	0.34 *	0.00	0.12	0.28
P3 × P8 (W996 × Bani-Suef-7)	−0.12	0.37	3.14 *	1.91	0.27 *	0.06	0.06	−0.29
P4 × P5 (W1011 × W1518)	0.04	−1.49	−0.39	−0.76	−0.09	−0.19	−0.15	−0.30
P4 × P6 (W1011 × W1520)	−0.26	0.07	1.41	1.01	0.25	0.12	−0.14	0.27
P4 × P7 (W1011 × Bani-Suef-5)	−1.29	−0.53	3.48 *	2.91 *	−0.08	−0.20	0.45	−0.46
P4 × P8 (W1011 × Bani-Suef-7)	−3.12 **	−2.93 **	1.21	1.84	−0.01	−0.61 **	−1.71 **	−0.22
P5 × P6 (W1518 × W1520)	1.28	−0.69	−1.09	−1.16	−0.25	−0.49 **	−0.57	−0.36
P5 × P7 (W1518 × Bani-Suef-5)	−1.09	0.37	0.31	0.74	0.39 **	0.60 **	−0.41	−0.31
P5 × P8 (W1518 × Bani-Suef-7)	4.08 **	5.64 **	2.71	−1.33	0.72 **	0.85 **	0.16	−0.84
P6 × P7 (W1520 × Bani-Suef-5)	0.61	−0.06	−1.22	−2.49	−0.01	0.14	−0.04	−0.08
P6 × P8 (W1520 × Bani-Suef-7)	−0.56	−2.79 **	−0.16	0.11	0.26	0.09	−0.50	−0.95 *
P7 × P8 (Bani-Suef-5 × Bani-Suef-8)	−0.26	−1.39	−1.42	−0.99	−0.60 **	−0.19	−0.54	−0.70
LSD 5% (sij)	1.80	1.84	3.09	2.55	0.26	0.28	0.91	0.87
LSD 1% (sij)	2.39	2.45	4.10	3.39	0.34	0.38	1.21	1.16
LSD 5% (sij-sik)	2.66	2.73	4.57	3.78	0.38	0.42	1.35	1.29
LSD 1% (sij-sik)	3.53	3.62	6.06	5.01	0.51	0.56	1.79	1.71
LSD 5% (sij-skl)	2.51	2.57	4.31	3.56	0.36	0.40	1.27	1.22
LSD 1% (sij-skl)	3.33	3.41	5.72	4.72	0.48	0.53	1.69	1.62
**Cross**	**No. of Spikes** **/Plant**		**No. of Grains** **Per Spike**		**1000-Grain** **Weight (g)**		**Grain Yield** **Per Plant (g)**	
	**Well-W.**	**Drought-S.**	**Well-W.**	**Drought-S.**	**Well-W.**	**Drought-S.**	**Well-W.**	**Drought-S.**
P1 × P2 (W988 × W994)	−0.01	0.92	8.81 **	6.20 **	2.79	3.41 *	8.60 **	7.55 **
P1 × P3 (W988 × W996)	1.89	1.72 *	−4.15	−4.10 *	0.86	1.84	−0.14	−1.15
P1 × P4 (W988 × W1011)	−2.14	−1.75 *	−8.09 **	−6.03 **	−0.34	−1.06	−7.37 **	−6.95 **
P1 × P5 (W988 × W1518)	−0.38	0.42	−13.32 **	−9.66 **	−1.01	0.11	1.80 *	5.08 **
P1 × P6 (W988 × W1520)	−0.08	−0.08	−0.19	−4.20 *	2.09	0.68	8.66 **	2.55 **
P1 × P7 (W988 × Bani-Suef-5)	1.19	−1.21	4.35	2.67	0.56	−1.56	5.60 **	3.71 **
P1 × P8 (W988 × Bani-Suef-7)	3.36 **	1.89 *	4.15	3.44	3.59	1.54	4.00 **	1.71 *
P2 × P3 (W994 × W996)	−1.88	−2.08 *	−0.89	−0.06	1.59	2.74	2.03 *	−1.02
P2 × P4 (W994 × W1011)	1.76	1.79 *	−1.15	1.34	−2.61	−2.82	−0.87	−3.15 **
P2 × P5 (W994 × W1518)	−0.14	−0.71	−3.72	−2.63	0.39	−0.99	0.96	−0.79
P2 × P6 (W994 × W1520)	1.16	1.45	−1.25	−1.50	3.16	3.24 *	−0.17	1.01
P2 × P7 (W994 × Bani-Suef-5)	0.42	−0.35	3.61	2.04	3.96 *	3.68 *	2.10 *	3.85 **
P2 × P8 (W994 × Bani-Suef-7)	0.26	0.09	1.75	1.47	−0.34	0.11	1.83 *	2.51 **
P3 × P4 (W996 × W1011)	−0.01	0.59	−5.12 *	−5.30 **	1.13	0.94	1.06	0.15
P3 × P5 (W996 × W1518)	2.42	2.09*	4.98 *	8.07 **	1.46	0.44	4.90 **	6.18 **
P3 × P6 (W996 × W1520)	0.39	0.25	5.78 *	5.87 **	2.23	1.01	5.10 **	1.65 *
P3 × P7 (W996 × Bani-Suef-5)	1.32	0.45	2.65	2.40	2.03	2.11	2.03 *	1.48 *
P3 × P8 (W996 × Bani-Suef-7)	−0.51	0.22	3.78	2.50	0.06	0.54	1.10	−0.52
P4 × P5 (W1011 × W1518)	4.06 **	2.29 **	3.71	4.47 *	3.26	2.88	4.66 **	3.38 **
P4 × P6 (W1011 × W1520)	0.69	0.79	0.85	3.94 *	3.36	3.78 *	8.20 **	10.18 **
P4 × P7 (W1011 × Bani-Suef-5)	0.96	0.99	10.05 **	7.47 **	3.49	4.54 **	4.80 **	3.01 **
P4 × P8 (W1011 × Bani-Suef-7)	0.79	1.09	5.85 *	2.90	1.19	0.98	4.53 **	2.68 **
P5 × P6 (W1518 × W1520)	−0.21	−0.05	−1.39	−5.03 **	−1.97	−3.39 *	−8.30 **	−6.45 **
P5 × P7 (W1518 × Bani-Suef-5)	0.06	0.49	2.48	3.84 *	4.49 *	4.71 **	5.63 **	3.38 **
P5 × P8 (W1518 × Bani-Suef-7)	2.89 *	1.92 *	10.95 **	5.27 **	0.86	2.14	5.36 **	5.05 **
P6 × P7 (W1520 × Bani-Suef-5)	−0.31	1.32	−0.05	2.97	−3.74 *	−4.39 **	1.83 *	−0.15
P6 × P8 (W1520 × Bani-Suef-7)	1.52	0.75	2.75	5.07 **	−1.37	0.71	4.90 **	5.18 **
P7 × P8 (Bani-Suef-5 × Bani-Suef-8)	1.12	1.29	−5.39 *	−1.40	0.43	0.81	4.17 **	5.35 **
LSD 5% (sij)	2.39	1.71	4.83	3.52	3.73	2.92	1.78	1.38
LSD 1% (sij)	3.17	2.26	6.41	4.67	4.95	3.88	2.37	1.84
LSD 5% (sij-sik)	3.53	2.52	7.15	5.21	5.52	4.33	2.64	2.05
LSD 1% (sij-sik)	4.69	3.35	9.49	6.91	7.32	5.74	3.50	2.72
LSD 5% (sij-skl)	3.33	2.38	6.74	4.91	5.20	4.08	2.49	1.93
LSD 1% (sij-skl)	4.42	3.16	8.94	6.52	6.90	5.41	3.30	2.56

* and ** imply significance at 0.05 and 0.01 levels of probability, in the same order.

## Data Availability

The data presented in this study are available upon request from the corresponding author.

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
