# Peer review of "Molecular Genetic Diversity of Local and Exotic Durum Wheat Genotypes and Their Combining Ability for Agronomic Traits under Water Deficit and Well-Watered Conditions"

_life, 2023, doi:10.3390/life13122293_

Round 1

Reviewer 1 Report

Comments and Suggestions for Authors

life-2695945 provides some valuable information to the researchers and readers. The subject of the manuscript is consistent with the scope of the Journal. I suggested that the manuscript need to be major revised before it is accepted by this journal.

1.Manuscript needs through language editing.

2.The logic and neat of introduction need to be further improved.

3.Line 51: ‘[1]’ should be superscript. Please check the full text.

4.Line 195: ‘p’ should be italic. Please check the full text.

5.The format of the subscript should be should be uniform, such as line 186, 187, 189 ,etc. Please check the full text.

6.The format of the figure needs to be adjusted, such as figure 5.

7.In the conclusions,in addition to summarizing the actions taken and results,please strengthen the explanation of their significance.

Comments on the Quality of English Language

Manuscript needs through language editing.

Author Response

Dear Editor,

We would like to thank you and the reviewers for the time and efforts devoted to our manuscript entitled “Exploring Combining Ability and Molecular Diversity of Local and Exotic Triticum durum Genotypes under Water Deficit and Well-watered Conditions” (Life-2695945). We have revised the manuscript according to the comments and suggestions pointed out by the reviewers. We have addressed the comments of the reviewers in a point-by-point below in red colour; in addition, we have highlighted all the associated changes made to the manuscript using track changes.

Yours sincerely,

Authors

Responses to Reviewers' Comments

Reviewer 1:

life-2695945 provides some valuable information to the researchers and readers. The subject of the manuscript is consistent with the scope of the Journal. I suggested that the manuscript need to be major revised before it is accepted by this journal.

Re: We would like to thank the Reviewer for his time dedicated to our manuscript and his positive assessment of our work.

  1. Manuscript needs through language editing.

Re: The manuscript has been carefully revised and the language has been improved.

  1. The logic and neat of introduction need to be further improved.

Re: The introduction has been revised and improved as requested (the changes were highlighted using track changes).

  1. Line 51: ‘[1]’ should be superscript. Please check the full text.

Re: The citation format has been revised to follow the journal style.

  1. Line 159: ‘P’ should be italic. Please check the full text.

Re: Done (line 175) as well as has been checked throughout the manuscript (lines 257 and 260).

  1. The format of the subscript should be should be uniform, such as line 186, 187, 189 ,etc. Please check the full text.

Re: The numbers with P have been uniform throughout the text.

  1. The format of the figure needs to be adjusted, such as figure 5.

Re: The figures have been improved as suggested.

  1. In the conclusions, in addition to summarizing the actions taken and results please strengthen the explanation of their significance.

Re: The conclusion has been revised and improved as requested (please see the conclusion in the revised version).

Thanks so much for your review which contributed considerably to improve our manuscript.

Reviewer 2 Report

Comments and Suggestions for Authors

Manuscript ID: life-2695945

Title: Exploring Combining Ability and Molecular Diversity of Local and Exotic Triticum durum Genotypes under Water Deficit and Well-watered Conditions.

Authors: Mohamed EL-Hity, Fatmah A. Safhi, Mohamed M. Kamara, Ahmed Galal, Eman Gamal El-Din, Medhat Rehan, Mona A Farid, Said Behiry, Mohamed El-Soda, and Elsayed Mansour.

The manuscript contains novel information which can further strengthen the existing knowledge of the field, particularly Triticum durum crop productivity with respect to water deficit conditions. Scientists planned their study according to need of time. Moreover, the research area seems well motivated; data analyses are technically correct and research findings support the claim and objectives properly made in the manuscript. Results are correctly presented and compared with existing knowledge of the field, and I think results have some potential broader applicability. Please consider addressing the following concerns and incorporate suggestions before any consideration to publish the work.

Concerns

1.     Abstract looks very lengthy. Interpret the significant findings in abstract.

2.     Arrange the keywords alphabetically.

3.     Arrange your introduction section in a systematic scientific way and identify the study gap and provide a robust hypothesis. Try to avoid stating general information and be specific. Rewrite the objective ii.

4.     The materials and methods section also needs extensive improvement. “The first treatment (well-watered conditions) was irrigated five times throughout the whole season with a total amount of approximately 4400 m3/ha”, make it sure that water quantity is correct.

5.     Directly state the results with significant findings. Make the results section concise and specific. This section needs extensive revision. Figure quality may be improved. Critical values are missing. Two different colors of bars are not explained in the figure.

6.     Try to discuss results with recent literature and provide reasoning of the responses recorded. Improve the discussion with logical and reasoning approaches.

Conclusion must be short, specific, and quantified. Conclusion given here is too lengthy.

Comments on the Quality of English Language

Moderate editing of English language required.

Author Response

Reviewer 2:

The manuscript contains novel information which can further strengthen the existing knowledge of the field, particularly Triticum durum crop productivity with respect to water deficit conditions. Scientists planned their study according to need of time. Moreover, the research area seems well motivated; data analyses are technically correct and research findings support the claim and objectives properly made in the manuscript. Results are correctly presented and compared with existing knowledge of the field, and I think results have some potential broader applicability. Please consider addressing the following concerns and incorporate suggestions before any consideration to publish the work.

Re: We would like to thank the Reviewer for his time dedicated to our manuscript and his positive assessment of our work.

Concerns

  1. Abstract looks very lengthy. Interpret the significant findings in abstract.

Re: The abstract has been revised and more details have been added as suggested (please see the abstract in the revised version)

  1. Arrange the keywords alphabetically.

Re: The keywords have been arranged alphabetically as requested.

  1. Arrange your introduction section in a systematic scientific way and identify the study gap and provide a robust hypothesis. Try to avoid stating general information and be specific. Rewrite the objective ii.

Re: The introduction has been carefully revised, and the hypothesis has been provided, the objectives have been revised as requested (the changes were highlighted using track changes).

  1. The materials and methods section also needs extensive improvement. “The first treatment (well-watered conditions) was irrigated five times throughout the whole season with a total amount of 4400 m3/ha”, make it sure that water quantity is correct.

Re: The materials and methods section has been carefully revised, the water amount is correct.

  1. Directly state the results with significant findings. Make the results section concise and specific. This section needs extensive revision. Figure quality may be improved. Critical values are missing. Two different colors of bars are not explained in the figure.

Re: The results have been revised; the quality of figures has been improved. The critical values are presented as bars on the top of the columns corresponding with LSD (P<0.05). The colors of the bars have been explained in Figures 2 and 3 by legends in Figures 2A and 3A.

  1. Try to discuss results with recent literature and provide reasoning of the responses recorded. Improve the discussion with logical and reasoning approaches.

Re: The discussion has been revised and improved as requested (please see the discussion in the revised version)

Conclusion must be short, specific, and quantified. Conclusion given here is too lengthy.

Re: The conclusion has been revised, shortened, and specified as requested (please see the conclusion in the revised version)

Moderate editing of English language required.

Re: The manuscript has been carefully revised and the language has been improved.

Thanks so much for your review which contributed considerably to improve our manuscript.

Reviewer 3 Report

Comments and Suggestions for Authors

The article "Exploring Combining Ability and Molecular Diversity of Local and Exotic Triticum durum Genotypes under Water Deficit and Well-watered Conditions" was received. Nevertheless, some corrections are required.

1. The title of the manuscript should be revised.

2. Abstract: The problem, methods, conclusions, and implications of the article should all be concisely presented, which currently needs to be improved. The abstract is too descriptive (check the word limit). Please start it with the objective of the paper and make the abstract more concise. 

3. The introduction is hard to follow; it needs to be better focused. Please improve it with the help of the latest references to justify the paper's novelty. 

4. Materials and methods are insufficient. Line 142 "ten main"?Specify it. Data collection (Line 141-147) should be more brief with reference and when it was measured (the observation stage should be clearly mentioned). Some paragraphs are improper (Line 148-156). Statistical analysis: Which software was used for combining ability analysis and Principal Component Analysis, as it is not provided in the MS. 

5. The results could be presented better. The research results should be stated more engagingly and logically to captivate readers and successfully communicate the findings. The presentation of results can be substantially improved by using clear, succinct language and logically arranging the information. 

6. It is essential to include more relevant citations in the discussion section to support your findings. Your results will have more validity and credibility if you include recent research findings. Referencing (citations) are not proper in throughout the manuscript please include in-text citations to support specific claims and statements. 

7. References are not formatted according to journal guidelines

Comments on the Quality of English Language

The language of the manuscript doesn't meet the minimum standards of publication. Many sentences are hard to understand. The author should pay more attention to the usage of punctuation. 

Author Response

Reviewer 3:

The article "Exploring Combining Ability and Molecular Diversity of Local and Exotic Triticum durum Genotypes under Water Deficit and Well-watered Conditions" was received. Nevertheless, some corrections are required.

Re: We would like to thank the Reviewer for his time dedicated to our manuscript.

  1. The title of the manuscript should be revised.

Re: The title has been revised and modified as requested to be “Molecular Genetic Diversity and Combining Ability of Local and Exotic Durum Wheat Genotypes under Water Deficit and Well-watered Conditions”

  1. Abstract: The problem, methods, conclusions, and implications of the article should all be concisely presented, which currently needs to be improved. The abstract is too descriptive (check the word limit). Please start it with the objective of the paper and make the abstract more concise.

Re: The abstract has been revised and improved as requested (please see the abstract in  the revised version)

  1. The introduction is hard to follow; it needs to be better focused. Please improve it with the help of the latest references to justify the paper's novelty.

Re: The introduction has been revised and improved as requested (the changes were highlighted using track changes).

  1. Materials and methods are insufficient. Line 142 "ten main"?Specify it. Data collection (Line 141-147) should be more brief with reference and when it was measured (the observation stage should be clearly mentioned). Some paragraphs are improper (Line 148-156). Statistical analysis: Which software was used for combining ability analysis and Principal Component Analysis, as it is not provided in the MS.

Re: The section of materials and methods has been revised as requested. "ten main" has been modified to be “were measured from ten spikes were randomly harvested from each plot” (line 157). More details and references have been added to the section of “2.3. Data Collection” (lines 153-160). Drought tolerance indices have been modified as suggested (lines 162-172). The used software has been added as suggested (lines 186-187).

  1. The results could be presented better. The research results should be stated more engagingly and logically to captivate readers and successfully communicate the findings. The presentation of results can be substantially improved by using clear, succinct language and logically arranging the information.

Re: The results have been revised and improved as requested.

  1. It is essential to include more relevant citations in the discussion section to support your findings. Your results will have more validity and credibility if you include recent research findings. Referencing (citations) are not proper in throughout the manuscript please include in-text citations to support specific claims and statements.

Re: The discussion has been revised, more recent and relevant citations have been added throughout the manuscript.

  1. References are not formatted according to journal guidelines

Re: The reference format has been modified to follow the journal style.

The language of the manuscript doesn't meet the minimum standards of publication. Many sentences are hard to understand. The author should pay more attention to the usage of punctuation.

Re: The manuscript has been carefully revised and the language has been improved.

Thanks so much for your review which contributed considerably to improve our manuscript.

Reviewer 4 Report

Comments and Suggestions for Authors

Here are my comments:

- The rationale for selecting the specific parental lines and SSR markers needs to be clarified. Why were these materials chosen for the study? 

- More details are needed on the field experimental design, plot layout, replications, etc. This is important for evaluating the results.

- The combining ability analysis methods require more explanation for readers less familiar with this approach. Define GCA, SCA, and provide context on interpreting GCA/SCA ratios. 

- The drought tolerance index formulas should be defined in the materials and methods section rather than only in the results. This will improve clarity.

- The presentation of results lacks sufficient context at times. For example, the genetic distance data between parents could be compared/contrasted to published literature. 

- The discussion of results is mainly descriptive in parts. More interpretation and critical analysis would strengthen it. For instance, relating the findings on gene action to implications for breeding.

- The quality of some tables and figures could be improved. For example, enlarging the text and axis labels in the biplot figure, and proofreading the heatmap legend. 

- The conclusions highlight the main findings but could provide more insight into applications and future research directions.

- Carefully proofread the manuscript to identify any additional sections needing clarification or zones of disjointed logic from one paragraph to the next. 

Author Response

Reviewer 4:

Here are my comments:

Re: We would like to thank the Reviewer for his time dedicated to our manuscript.

- The rationale for selecting the specific parental lines and SSR markers needs to be clarified. Why were these materials chosen for the study?

Re: The parental genotypes were identified based on their tolerance to water deficit from a prior screening trial including diverse 25 durum wheat genotypes during the 2018-2019 winter season. The applied SSR markers associated with drought tolerance from earlier studies. More details and references have been added to the section of materials and methods (lines 111-118).

- More details are needed on the field experimental design, plot layout, replications, etc. This is important for evaluating the results.

Re: More details on the applied experimental design, number of replications, and plot layout have been added as requested (please see lines 146-149).

- The combining ability analysis methods require more explanation for readers less familiar with this approach. Define GCA, SCA, and provide context on interpreting GCA/SCA ratios.

Re: More details have been added on the applied method for analyzing combining ability and interpretation GCA/SCA ratio (please see lines 176-181).

- The drought tolerance index formulas should be defined in the materials and methods section rather than only in the results. This will improve clarity.

Re: The formulas of drought tolerance indices have been defined in the materials and methods as requested (lines 162-172)

- The presentation of results lacks sufficient context at times. For example, the genetic distance data between parents could be compared/contrasted to published literature.

Re: The section of results has been revised and improved. The genetic distance among assessed parents has been discussed based on published literature in the section of discussion (the changes were highlighted using track changes).

- The discussion of results is mainly descriptive in parts. More interpretation and critical analysis would strengthen it. For instance, relating the findings on gene action to implications for breeding.

Re: The discussion has been revised and improved as requested (please see the discussion in the revised manuscript)

- The quality of some tables and figures could be improved. For example, enlarging the text and axis labels in the biplot figure, and proofreading the heatmap legend.

Re: The quality of figures has been improved as requested.

- The conclusions highlight the main findings but could provide more insight into applications and future research directions.

Re: The conclusion has been revised and improved as suggested (please see the conclusion in the revised manuscript)

- Carefully proofread the manuscript to identify any additional sections needing clarification or zones of disjointed logic from one paragraph to the next.

Re: The manuscript has been carefully revised and improved. Thanks so much for your review which contributed considerably to improve our manuscript.

Round 2

Reviewer 1 Report

Comments and Suggestions for Authors

I have no comments.

Comments on the Quality of English Language

I have no comments.

Author Response

Reviewer 1:

I have no comments.

Re: Thanks so much for your review which contributed considerably to improve our manuscript.
The manuscript has been revised for a second round and improved as suggested.

Reviewer 2 Report

Comments and Suggestions for Authors

N/A

Comments on the Quality of English Language

Minor editing.

Author Response

Reviewer 2:

Minor English editing

Re: Thanks so much for your review which contributed considerably to improve our manuscript.

The manuscript has been revised for a second round and improved as suggested.

Reviewer 3 Report

Comments and Suggestions for Authors

Manuscript can be accepted

Author Response

Reviewer 3:

Manuscript can be accepted

Re: Thanks so much for your review which contributed considerably to improve our manuscript.